# Interactions Among Factors Influencing Product Innovation and Innovation Behaviour: Market Orientation, Managerial Ties, and Government Support

**Natenapang Thongsri [1] and Alex Kung-Hsiung Chang [2],***

[1]  Department of Tropical Agriculture and International Cooperation,
     National Pingtung University of Science and Technology, Pingtung 9120, Taiwan; nate_1218@hotmail.com
[2]  Department of Business Administration, National Pingtung University of Science and Technology,
     Pingtung 91201, Taiwan
*  Correspondence: bear419@mail.npust.edu.tw; Tel.: +886-8-7740374

**Abstract:** Ongoing globalization and changing customer needs make it increasingly difficult for firms to survive in the long term. Innovation is considered an important tool for firms in this environment. In particular, a firm's ability to cultivate innovative behaviour and implement product innovation for sustainability is important. This study explores resources and capabilities to enhance firm innovation behaviour and implementation of sustainable product innovation. The results provide insights on how firms can manage strategies for future sustainable innovations. We used a sample of 645 small- and medium-sized enterprises and presented the conceptual framework according to a resource-based view and relational capital. We specified three independent factors that enhance sustainable innovation and superior performance: market orientation, managerial ties, and government support. We used a questionnaire survey and structural equation modelling to evaluate the conceptual model. We found that interactions between business ties, customers, and competitor orientation can enhance sustainable product innovation, whereas interactions between government support and political relations can enhance the sustainability of innovation behaviour. Moreover, product innovation and innovation behaviour are mediators that can lead to superior firm performance. The results suggest ways entrepreneurs and public policy makers can promote sustainable innovation.

**Keywords:** product innovation; innovation behaviour; market orientation; managerial ties; government support

## 1. Introduction

A changing external environment, varying customer characteristics, and more sophisticated customer needs have made it increasingly hard for firms to retain competitiveness. Some entrepreneurs have struggled to compete and succeeded, while others have failed. Therefore, entrepreneurs and firms should develop effective strategies to cope in the current environment [1–3]. Innovation is an important weapon that firms can use to safeguard their existence, and it can give firms a competitive advantage over rivals. Innovation involves firms altering existing methods or creating new methods related to the provision of products and services. It helps firms deliver economic and social value, enter new markets, and expand existing markets to amplify performance [4]. Innovation is an essential complement for firms, it leads to sustainable competitive advantages and superior performance [5–8]. Hence, to ensure superior performance, firms need to consider resources and capabilities that can enhance

sustainable innovation development. A firm's responsibility is to cultivate and integrate internal and external resources to sustain innovation excellence [9]. This can take the form of sustainable innovation activities: developing new products, updating existing products, and shaping employee behaviour [10]. These activities can create unique and difficult-to-imitate advantages [11]. However, previous studies indicate that innovative operation of small- and medium-sized firms can be a huge challenge for entrepreneurs [12,13], who may face barriers in accessing finance and lack necessary business information [14–16]. Therefore, in our study, we focus on small and medium-sized enterprises (SMEs). These are creative businesses with the need to develop different strategies through investment in sustainable innovation to meet the demands of changing markets.

In today's business environment, firms should realize that innovation is a significant strategy [17] and that improving strategies is vital for firms that intend to build superior forms of sustainable innovation [10]. Firms must strive to keep their products current and attractive to the market, create more value from existing product lines [18], and encourage innovative work among personnel [19], taking advantage of creativity and unique skills among employees that are difficult for other firms to imitate [20]. Previous research has suggested that to increase innovation capabilities, a firm must cultivate both tangible and intangible assets (firm-specific factors) [21] to ensure that sustainable innovation is relevant for addressing market needs and business aims and for securing competitive advantage. A review of the literature suggests that three critical firm resources and capabilities are associated with sustainable innovation development and superior performance. Firstly, a market orientation that comprises customer orientation, competitor orientation, and interfunctional coordination [22,23]. Secondly, firm managers' business and political ties play a crucial role [24]. Thirdly, support from the government sector is also important [25].

The strategic management of sustainable innovations within SMEs can be accomplished by enhancing relationships with external resources such as customers, value chain networks, governments or authorities, and knowledge networks [26]. This kind of market orientation can help firms understand rare resources in terms of customer needs [27]. Entrepreneurs should adopt a market orientation and consider their propensity to anticipate customers' future needs and then develop products and services accordingly [28]. In this way, a market orientation plays a vital role in supporting firm innovation [22,23]. Previous related research supports the idea that a market orientation can enhance firm performance [29–31]. Moreover, there is an existing argument that a market orientation is not directly linked to performance [32] but is instead mediated by innovation [33–35]. From a behavioural perspective, studies have described market orientation as an aspect of organizational culture [29]. SMEs have a unique organizational culture that is different from large firms, i.e., the corporate culture of SMEs tends to be more organic [36].

Many studies in creative business have suggested that government agencies are vital to firm competitiveness [37–41]. Governments can promote the upgrading of skills, human resources, or basic knowledge in firms, which can ease pressure on firms that have lost competitive advantage [2]. The government also has an important but indirect effect on the industry's competitive development, in that the government can act as an exogenous variable [2]. Political and business ties are also significant in encouraging innovation [25]. Political connections can have a considerable influence in a transition economy when they set the stage for positive relationships with officers, which can have a strong effect on market and firm performance [42]. Firm managers can cultivate direct, strong, long-term connections with political leaders or government officials. They should also build solid relationships with buyers, suppliers, and competitors because the influence of political ties will decline as time goes on [43]. However, sometimes political ties may not actually enhance performance. In the event of political conflicts or changes in government, political ties can create costs for an organization and may not actuate innovation activities [24,44]. While there have been studies on government, managerial ties, product innovation, innovation behaviour, and performance, a conclusive empirical study that has explored these factors simultaneously has not been conducted.

This conceptual model addresses the research gap and makes an essential contribution to the literature by simultaneously investigating all these factors as well as the relationships between them. Firstly, we examine the relationships between market orientation, business ties, and product innovation. Secondly, we expand the scope of previous research on the role of government and political relations in promoting sustainable innovation and superior market and financial performance. Thirdly, we examine product innovation and how innovation behaviour acts as a significant mediator to enhance firm performance. Furthermore, this study illustrates the relationship between managerial ties and relational capital [7,8,45]. It contributes to the further development of related theory through empirical research considering managerial ties as a moderator in the development of sustainable innovation.

This study employs a resource-based perspective to identify links between the conceptual framework constructs (see Figure 1). Our research also expands upon existing theory by empirically testing the proposed links between all factors. The remainder of the paper is structured as follows. Section 2 describes the theoretical foundation and conceptual framework. Section 3 illustrates our methodology. Section 4 shows the research results. Section 5 presents our conclusions, a discussion of the theoretical and managerial implications, a description of the study's limitations, and recommendations for future research.

## 2. Literature Review and Research Hypotheses

### 2.1. Resource-Based View and Sustainable Innovation

In the resource-based view, a business can sustain its competitive advantage compared with other firms by using its resources and capabilities [46], but these must be unique, valuable, inimitable, and difficult to substitute [47]. These firm-specific resources can present as tangible assets, intangible assets, and capabilities [48]. Firms have different types of resources that are unique and difficult to imitate. These resources can inform strategic action, giving rise to new ideas about how to improve firm performance [28]. When capitalizing on firm-specific resources, a firm will realize a strong push towards better performance [49–51]. To enhance competitive advantage and achieve organizational goals in highly competitive markets, firms need to use their capabilities to foster innovative behaviour in employees [20,52–54] and to inspire product innovation [1]. Firm-specific resources can help firms with sustainable innovation development particularly in the areas of relationships with markets, knowledge partners, and institutions supportive of sustainable innovation [55]. In sum, firms need to consider an entire range of factors when striving for excellence in sustainable innovation. These factors relate to employees, customers, and other stakeholders as well as to resources and management [10].

Innovation is a primary strategy of firms to enhance performance and competitive advantage [56]. In this study, we distinguish between two categories of innovation. The first is product innovation, in which firms actualize new ideas to create value [57]. Product innovation variables are indicators of the uniqueness of a product [58]. The second category is innovative work behaviour, which arises from human capital [19]. Examples of important innovative work behaviour are problem recognition and idea generation [59]. Some prerequisites for innovative behaviour are environmental features and diverse employee personalities [60]. Capable employees, who through their abilities, knowledge, skills, and developmental and creative ideas allow a firm to differentiate itself from its competitors [61], become key resources for organizations [62].

### 2.2. Market Orientation

Declining or unsuccessful firms may lack effective firm strategies and structures [63]. A market orientation may serve as a significant strategy for firms, because it can help firms perform activities in a way that increases value for buyers [29]. With a market orientation, firms can obtain accurate information about customers and competitors, respond promptly to new information, and disseminate information internally among various departments. Market orientation incorporates a customer and competitor orientation along with interfunctional coordination [64]. It focuses on customer needs that

are the most important to understand, the sophistication of buyers, efficient, existing competitors, and the interfunctional coordination of the firm [65].

Customer orientation involves the assessment of and appropriate response to customers' needs and desires [22], whereas a competitor orientation involves analysing and developing suitable responses to the activities of competitors [34]. When a firm understands its competitors, it is more likely to be able to develop products that are superior to others currently on the market (products offered by competitors or even by the company itself) [23]. Through customer and competitor orientation, a firm can promptly and effectively respond to market needs to offer superior customer value [66]. In addition, a firm needs to disseminate information internally among internal divisions and engage in interfunctional coordination. This ensures smooth integration and coordination of all departments in the firm [22] and creates communication linkages that reduce conflict and insecurity [43]. Information obtained about market factors and customer needs should be announced widely within an organization so that it can develop and implement new strategies [67]. These activities are essential so that the entire firm has full access to market needs and information [68].

### 2.3. Managerial Ties

Managerial ties can help firms cope with uncertainty in formal institutional systems and secure external resources [43]. Managerial ties fall into two categories: business ties, which are relationships with suppliers, buyers, competitors, and other stakeholders [69]; and political ties, which are relationships with political officials or government organizations [42]. These two kinds of network relationships are distinctly different and can provide unique kinds of strategic resources to firms [43].

### 2.4. Market Orientation, Business Ties, and Product Innovation

The product life cycle consists of four phases: introduction (development or design); growth; maturity; and decline [70]. In the current market, the environment (buyer demands and rivals) is causing product life cycles to become shorter. Thus, products designed for sustainability are better equipped for the market [58]. Creativity and development characterize the first stage of the product life cycle. In this stage, firms are tasked with building relationships with external stakeholders [71]. The relationships with suppliers, customers, and other stakeholders serve as a stimulus for innovation, and they promote sustainability [72]. Relational capital—whether in the form of relationships with customers or suppliers—is the foundation of the knowledge-sharing process that precedes the development of new knowledge [73]. The significance of business ties is that customers, suppliers, and competitors can provide information that incites a firm to develop new strategies to improve products and services [74]. In a dynamic market environment, customers move from being unknown agents to controlling engagement, communication, and collaboration with a company [3]. Firms can use the interaction between customer orientation and business networks to better understand and respond to the needs of customers [75]. Meanwhile, a competitor orientation may attract imitation and hinder creativity and innovation [22]. When firms emphasize competition, they might use business ties to gather information on competitors' strengths, weaknesses, cost savings, or operational changes, but they may not invest sufficiently in radical innovation [24]. A firm often imitates competitors' policies and operations after obtaining competitor information [76]. When a firm receives information from customers, competitors, and suppliers, disseminating information to other organizations is essential [43]. Employee participation and communication in the form of sharing information with other employees improve interfunctional coordination of the firm [77]. Moreover, positive network relationships are imperative for achieving a higher degree of product creativity [7,56,78]. Therefore, significant interactions exist between a market orientation and business ties that can enhance the firm's sustainable product innovation. Thus, we suppose the following:

**Hypothesis 1 (H1).** *Business ties interact with a customer orientation and can enhance the effect on product innovation (H1a); Business ties interact with a competitor orientation and can enhance the effect on product innovation (H1b); Business ties interact with interfunctional coordination and can enhance the effect on product innovation (H1c).*

*2.5. Government Support, Political Ties, and Innovation Behaviour*

State institutions can bolster firms' innovation activities by supporting knowledge diffusion, technology transfer, funding searches, and project management [79]. Thus, raising the firm's relationships with institutions is crucial for promoting sustainable innovation development. Political relationships include close ties with institutions such as industrial bureaus or regulatory agencies, banks, and tax bureaus [42]. When firms have strong institutional networks, they can more easily gain access to critical external resources, accurate and timely information, and legal protection [24,25]. For example, when managers have positive relationships with government officials, they can obtain useful information on industrial regulations and policies [80]. Governments, whether local or national, can also have an impact on firm expansion by providing firms with opportunities to advance their learning [81]. A key governmental priority is its investment in innovation, which means investment in human and creative capital [82]. This is because a government often serves as the principal engine for progress in educational systems, basic infrastructure, and research [2]. Governments launch measures to help firms develop. They support skill development through education and training, promote entrepreneurship, and provide access to financing [83,84]. They can establish training centres to allow for knowledge exchange between firms, facilitating collaboration and creating opportunities for firms to work together to find innovative solutions to challenges within industries [85]. When government support is interrupted, a firm with strong political ties can obtain assistance from government officials [44]. In sum, a firm should cultivate strong ties with government officials and institutions to ensure timely access to accurate information as well as to ensure access to external knowledge that can support innovation for sustainability. Accordingly, we set the following assumptions:

**Hypothesis 2 (H2).** *Political ties have a positive moderating effect on the relationship between government support and innovation behaviour.*

*2.6. Market Orientation, Product Innovation, and Firm Performance*

A market orientation involves gathering market information from customers and competitors. It also encourages interfunctional coordination and the sharing of information and resources. Firms should understand the needs of target customers by discussing competitors' strengths, weaknesses, and strategies and by using internal communication about new ideas to create value-added and unique products. [29,86,87]. These activities can help build and sustain superior performance. However, some research has suggested that further analysis of the effect of market orientation on firm performance is needed, and an argument exists that market orientation cannot directly influence firm performance. In other words, studies should explore mediating factors between market orientation and performance [31,32,65]. Some studies have suggested that the resources and capabilities of a firm can serve as mediator variables [31,65].

When consumer demand is changing and growing, firms are more likely to invest in new products or modify original products, which affects overall corporate success [88]. Innovative products are essential for companies who want to meet consumer needs and feed the appetites of consumers for new technology [58]. Product innovation is a firm resource that can act as a mediator variable between marketing strategies and firm performance [50]. Customers, suppliers, and other relevant stakeholders help firms to develop sustainable product innovation by supplying new ideas, knowledge, demands, and requirements [89,90]. Therefore, we propose that product innovation is a significant mediator variable between market orientation and firm performance.

**Hypothesis 3 (H3).** *Product innovation mediates the relationships between customer orientation and firm performance (H3a); Product innovation mediates the relationships between competitor orientation and firm performance (H3b); Product innovation mediates the relationships between interfunctional coordination and firm performance (H3c).*

*2.7. Government Support, Innovation Behaviour, and Firm Performance*

As we discussed earlier, institutional support plays a vital role for firms by ensuring access to rare resources, funding, financing, and project support [66]. Porter suggested that the government is an important exogenous variable; in other words, governments do not control competition but can influence it [2]. Human development is considered a central goal of governmental economic growth policies. Local and state governments can enact policies that influence human resources and knowledge [2] through training schemes that support employee participation in the development of sustainable business [26]. Therefore, our study considers innovation behaviour as a mediating variable between government support and sustainable implementation. The innovative behaviour of employees may help firms to obtain economic benefits that differentiate it from competitors [91]. This occurs when employees generate new ideas and knowledge or take the initiative to develop a new product; these activities can enhance both market and financial performance [92]. The experience and knowledge of personnel can affect firm performance [93] and cannot easily be imitated by other firms [94]. Thus, firms should encourage knowledge sharing as well as value and develop their workforce to maintain their competitive advantage. This will increase the potential value of their human capital pool [95]. Therefore, we predict the following.

**Hypothesis 4 (H4).** *Innovation behaviour is a mediator between government support and firm performance.*

*2.8. Government Support, Innovation Behaviour, and Product Innovation*

Government plays both direct and indirect roles in the implementation of sustainable innovations by firms; nevertheless, it can still act as a barrier by enacting rules and regulations [1,2,70]. Governments can establish policies to suit SMEs by considering their needs to develop sustainable innovation activities [96,97]. Governments can also support sustainable innovation for these firms by providing initial capital and full support or by offering non-monetary subsidies, such as places for knowledge exchange, information, patents, and R&D activities [85,98]. Government support is critical to sustainable innovation development because the market alone could not provide sufficient incentives for the development of the knowledge necessary to enhance the sustainability of products [96]. Governments can support creative individuals through education and training [98], and employee training and R&D are essential for sustainable product innovation [55]. Generally, firms develop their employees before investing in other areas of the organization, as employees are soft aspects of innovation [10]. Employees demonstrating innovation behaviour often offer ideas and participate in their implementation, and this innovative behaviour can enhance the creative outputs anticipated by a firm [19]. Employees accumulate creativity, experience, knowledge, and abilities that can be used in product development [74]. Thus, a firm's innovation behaviour acts as a mediator to enhance the sustainability of products. We hypothesize the following:

**Hypothesis 5 (H5).** *Innovation behaviour is a mediator between government support and product innovation.*

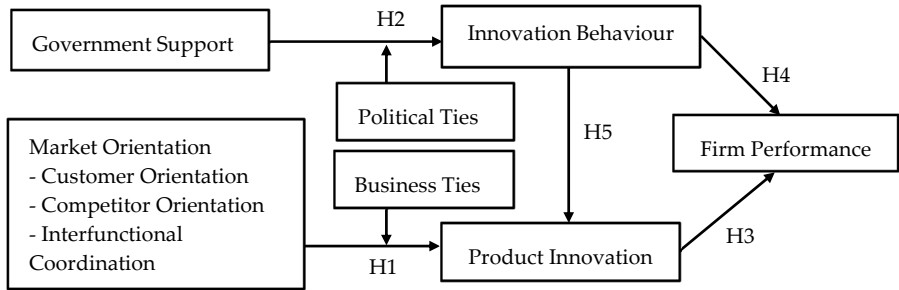

**Figure 1.** The study's conceptual framework

## 3. Material and Methods

### 3.1. Sampling and Data Collection

Our objective was to gain a deeper understanding of the potential for a firm's dynamic resources and capabilities to enhance innovation activities and performance. To test all hypotheses, we investigated SMEs in the creative sectors of six regions in Thailand (northern, western, eastern, northeast, southern, and central). We separated the questionnaire into two sections. The first part collected information about firm characteristics, including the type of business, number of employees, and number of years since the firm was established. The second part contained items related to measuring the independent variables of market orientation, managerial ties, and government support, and the dependent variables of product innovation, innovation behaviour, and firm performance. In the first step, we constructed an English version of the questionnaire and discussed each question with a bilingual specialist. We ensured the survey's coincident meaning as well as its content validity and reliability by translating it into Thai and then back-translating it into English. We executed five interviews with administrators from related and supporting SMEs as well as with five owners of SMEs; we asked them to evaluate the questions' sufficiency and completeness. Their responses guided us to adjust the questionnaire to enhance entrepreneurial clarity. Then, we asked a sample of 30 SME entrepreneurs to complete the questionnaire and provide recommendations to improve its style and address any weaknesses. An analysis of this pre-test survey indicated that the respondents clearly understood all questions. The Cronbach's alpha values for all items were well above the commonly acceptable value of 0.7 [99]; therefore, the elementary data were considered internally stable and consistent.

We then selected a sample of 670 SMEs from those enrolled with the Office of SME Promotion. A package was mailed to each SME's owner that included a letter of introduction and explanation, the questionnaire, and a return envelope with postage stamp. A follow-up was conducted by phone to elicit a response if none was received. The number of questionnaires increased by 30% from the sample size to compensate for non-responses. This was the amount required for the requisite level of confidence and precision [100,101]. Of the 870 questionnaires sent, 645 were completed, with a response rate of 96.26%.

### 3.2. Measures

Several item scales were used to evaluate the hypotheses. Multiple items were developed to measure market orientation, managerial ties, government support, innovation, and firm performance. Moreover, to specify the contexts of these variables and resulting firm performance, we used a five-point Likert scale for all measures.

This study measured five items of product innovation similar to the works of Akman and Yilmaz, Prifti and Alimehmeti, and Jansen et al. [22,87,102]. Four items of innovation behaviour were adopted from Liu [103]. We developed an 18-item scale from previous studies [22,64–66,104] to measure market orientation. We measured government support by developing five items from Zhang et al. and Sheng et al. [25,44]. Regarding managerial ties, we adopted variables from Peng and Luo [42]. Regarding firm performance, we developed a five-item measure based on the works of Cacciolatti and Lee [104].

The scale ranged from 1 (*very unsatisfactory*) to 5 (*very satisfactory*). Regarding each performance indicator, respondents were asked to characterize their product or service's development in terms of opportunities for sales ratio growth, return on investment, market share, consumer satisfaction, and consumer loyalty. To describe the effects of external variables, composite firm size and firm age were the control variables. We used the number of employees to specify company size and the number of years since the business was founded to specify firm age.

## 4. Results

### 4.1. Respondents' Characteristics

This study distributed 870 copies of the prorated questionnaire, and 645 complete samples were collected. The most responsive subsector was design, followed by crafts and fashion, with response rates of 24.8%, 21.9%, and 15.5%, respectively. Other business categories were as follows: food, visual art, printing, tourism, medicine, software, advertising, performing art, architecture, film, and music. Respondents' most popular business type was corporations, followed by sole proprietorships, and community enterprises, with rates of 42.2%, 31%, and 20.3%, respectively.

### 4.2. Confirmatory Factor Analysis and Exploratory Factor Analysis

We ensured the validity and the reliability of the questionnaires by performing an exploratory factor analysis (EFA) followed by confirmatory factor analysis (CFA). Firstly, we evaluated the validity and reliability of the studies' constructs using EFA and calculated Cronbach's alpha. Nine factors were extracted from 43 items; the total variance explained collectively by all factors was 78.76%, with Cronbach's alpha values ranging from 0.71 to 0.95, with all items surpassing 0.60 for an exploratory study [105]. We extracted factors for cases in which the eigenvalue was greater than 1.0. The 43 items had factor loading values ranging from 0.57 to 0.94, and all items were higher than the recommended threshold of 0.50. We analysed support for factors using principal component analysis and varimax rotation. The evaluation results indicated a significant Bartlett's test for sphericity, and the Kaiser-Meyer-Olkin value was 0.80. Thus, the data were appropriate for a factor analysis. Secondly, we conducted a CFA to estimate construct validity. As noted in Appendix A, each variable's item-total correlation was greater than 0.50, exceeding the recommended standard. The result of the model indicates a desirable fit to the data: cmin/df = 2.97; comparative fit index (CFI) = 0.96; Tucker-Lewis Index (TLI) = 0.94; normed fit index (NFI) = 0.94; goodness-of-fit index (GFI) = 0.88; adjusted goodness-of-fit index (AGFI) = 0.83; root mean square error of approximation (RSEMA) = 0.05 [106,107]. All items were significant ($p < 0.01$), and the standardized factor loadings of all items were 0.51 to 0.98. These results indicate convergent validity. Moreover, the results of the constructs' composite reliability (CR) ranged between 0.75 and 0.95, with values greater than the acceptable 0.70 levels. There was an average variance extracted (AVE) value between 0.51 and 0.80, which is higher than the acceptable level of 0.50. For discriminant validity, we checked the maximum variance value of pairs of constructs to verify that their values were not higher than AVE. The output indicates that each construct's highest variance value was lower than AVE [108]. Overall, these results indicate satisfactory reliability and validity. Table 1 shows the means, standard deviations, and correlations between all the study's constructs.

**Table 1.** Descriptive statistics and correlation matrix.

| Variable | 1 | 2 | 3 | 4 | 5 | 6 | 7 | 8 | 9 | 10 | 11 |
|---|---|---|---|---|---|---|---|---|---|---|---|
| 1. Customer orientation | 1.00 | | | | | | | | | | |
| 2. Competitor orientation | 0.17 | 1.00 | | | | | | | | | |
| 3. Interfunctional coordination | −0.01 | −0.04 | 1.00 | | | | | | | | |
| 4. Business ties | 0.15 ** | 028 ** | 0.05 | 1.00 | | | | | | | |
| 5. Political ties | 0.22 ** | 0.10 ** | −0.13 ** | 0.10 ** | 1.00 | | | | | | |
| 6. Innovation behaviour | 0.14 ** | 0.10 * | −0.11 ** | 0.14 ** | 0.20 ** | 1.00 | | | | | |
| 7. Product innovation | 0.14 ** | −.05 | −0.13 ** | −0.12 ** | 0.17 ** | 0.18 ** | 1.00 | | | | |
| 8. Government support | −0.03 | 0.10 * | −0.11 ** | 0.15 ** | 0.05 | 0.44 ** | 0.01 | 1.00 | | | |
| 9. Firm performance | 0.15 ** | 0.05 | −0.03 | 0.12 ** | 0.31 ** | 0.62 ** | 0.28 ** | 0.43 ** | 1.00 | | |
| 10. Firm size | 0.08 ** | −0.06 | 0.09 * | 0.09 * | −0.04 | −0.07 | 0.03 | −0.03 | 0.06 | 1.00 | |
| 11. Firm age | 0.13 ** | 0.10 * | −0.01 | 0.35 ** | −0.04 | −0.05 | −0.06 | 0.13 ** | 0.08 * | 0.13 ** | 1.00 |
| Mean | 3.88 | 3.77 | 3.63 | 4.00 | 3.12 | 3.09 | 3.71 | 2.94 | 2.68 | 1.36 | 2.27 |
| Standard deviation | 0.61 | 0.64 | 0.76 | 0.84 | 0.71 | 0.77 | 0.82 | 0.97 | 1.00 | 0.48 | 0.86 |

Sample size: 645. Significance level: ** $p < 0.01$, * $p < 0.05$.

### 4.3. Structural Equation Models and Relationship Testing

This study used structural equation models with the maximum likelihood method to ensure the validity of the conceptual framework illustrated in Figure 1, and we investigated the relationships between all constructs.

Table 2 illustrates the results of H1a–H1c and H2, concerning the nature of interactions between business ties and market orientation and between political relations and government support, respectively. These are all the interactions that affect innovation. The results are as follows. Model 1 shows the effect of firm size and firm age; model 2 consists of three elements of market orientation and moderator variables (business ties); and model 3 shows the moderating effect of business ties on the relativity of market orientation to product innovation. Model 3 and Figure 2a indicate a positive and significant interaction between business ties and customer orientation ($\beta = 0.14$; $p < 0.01$), which indicates that business ties strongly reinforce customer orientation to enhance product innovation. Figure 2a shows a simple slope analysis of the moderating effect of low and high business ties on the relationship between customer orientation and product innovation. The interaction plot indicates that customer orientation has a strong positive impact on product innovation when the firm has high-level business ties, whereas customer orientation has a negative effect on product innovation when the firm has low-level business ties. Therefore, H1a was supported. Conversely, the results show an adverse interaction effect of business ties and competitor orientation on innovation ($\beta = −0.15$, $p < 0.01$). This supports H1b. The interaction plot in Figure 2b indicates the reversed effect of business ties on the relationship between competitor orientation and product innovation. In other words, when the firm has strong business ties, the relationship between product innovation and competitor orientation is weakened; when the firm has weak business ties, the relationship between product innovation and competitor orientation is strengthened. The results show no significant interaction between business ties and interfunctional coordination. Therefore, H1c is not supported. Finally, the moderating effect of political ties and government support on innovation behaviour variables is confirmed in Table 2. Model 4 consists of control variables, model 5 includes government support, and model 6 indicates that political ties moderate the government support and innovation behaviour relationship ($\beta = 0.15$, $p < 0.01$). As such, H2 is supported. The interaction is presented in Figure 3; the interaction plot of the moderating effect of government support and innovation behaviour at high and low levels of political ties indicates that government support has a strong positive effect on innovation behaviour whenever the firm has high-level political ties; however, the effect of government support becomes less positive when the firm has low-level business ties.

Product innovation and innovative behaviour were predicted as mediators between market orientation and firm performance (H3a–H3c) and between government support and firm performance (H4). The result shown in Table 3 indicates a satisfactory fit of the model data (Cmin/df = 2.36; CFI = 0.95; TLI = 0.90; NFI = 0.91; GFI = 0.98; AGFI = 0.94; RSEMA = 0.05). After entering the effect of the mediator (product innovation), the effect of customer orientation on firm performance is not

significant (β = 0.01) and customer orientation is positively correlated with product innovation (β = 0.13, $p$ < 0.01). Moreover, product innovation positively affected performance (β = 0.17, $p$ < 0.01). In other words, customer orientation had an indirect effect on the performance of a firm; therefore, H3a is fully supported. Conversely, competitor orientation and firm performance had a non-significant relationship (β = −0.02) and competitor orientation has an insignificant effect on product innovation (β = −0.010). Thus, H3b is not supported. Furthermore, interfunctional coordination was significantly related to firm performance (β = 0.10, $p$ < 0.01), and it had a significant effect on product innovation (β = −0.10, $p$ < 0.01), while product innovation positively affected firm performance. Therefore, the finding that product innovation can partially mediate the correlation between interfunctional coordination and performance partially supports H3c. H4 relates to the underlying encouragement of government that can enhance a firm's operations through innovation behaviour. The result showed a significant positive effect between government support and performance (β = 0.23, $p$ < 0.01) when innovation behaviour acted as a mediator. Similarly, government support and innovation behaviour showed a positive effect (β = 0.40, $p$ < 0.01). In addition, innovation behaviour had a positive effect on firm performance (β = 0.44, $p$ < 0.01), so H4 is partially supported. Table 3 shows that support of the government had an indirect effect on product innovation. The government had an insignificant (β = −0.05) effect on product innovation when innovation behaviour entered into the model as a mediator, and government was positively related to innovation behaviour (β = 0.40, $p$ < 0.01). Also, innovation behaviour positively affected product innovation (β = 0.19, $p$ < 0.01), thereby fully supporting H5. Table 3 also presents the effect of control variables and shows an adverse impact of firm size on innovation behaviour. This is a significant liability to which SMEs need to pay more attention. However, there appeared to be a positive correlation between firm size and firm performance. In this case, it seems that SMEs enjoy superior performance and competitive advantage. Also, firm age positively affected innovation behaviour, with firm age as an advantage for firms to engage in innovation behaviour.

**Table 2.** Interaction between business ties, political ties, market orientation, and government support.

| Variable | Product Innovation | | | Innovation Behaviour | | |
|---|---|---|---|---|---|---|
| | Model 1 | Model 2 | Model 3 | Model 4 | Model 5 | Model 6 |
| Control Variable | | | | | | |
| Firm Size | 0.04 (0.98) | 0.04 (0.80) | 0.04 (0.89) | −0.09 * (−2.27) | −0.07 (−1.90) | −0.07 (−1.94) |
| Firm Age | −0.02 (−0.43) | −0.04 (−0.88) | −0.02 (−0.36) | 0.16 ** (4.23) | 0.11 ** (3.07) | 0.12 ** (3.55) |
| Moderator Variable | | | | | | |
| Business Ties (BT) | −0.12 ** (−2.85) | −0.12 ** (−2.90) | −0.12 ** (−2.85) | | | |
| Political Ties (PT) | | | | 0.20 ** (5.18) | 0.18 ** (5.06) | 0.11 ** (3.28) |
| Independent Variable | | | | | | |
| Customer Orientation (CUO) | | 0.14 ** (3.65) | 0.15 ** (3.85) | | | |
| Competitor Orientation (COO) | | −0.05 (−1.30) | 0.01 (0.18) | | | |
| Interfunctional Coordination (IFC) | | −0.13 ** (−3.38) | −0.12 ** (−3.18) | | | |
| Government Support (GS) | | | | | 0.41 ** (11.75) | 0.40 ** (11.62) |
| Interaction Variable | | | | | | |
| CUO*BT | | | 0.14 ** (3.78) | | | |
| COO*BT | | | −0.15 ** (−3.84) | | | |
| IFC*BT | | | −0.01 (−0.10) | | | |
| GS*PT | | | | | | 0.15 ** (4.68) |
| Model Statistics | | | | | | |
| $R^2$ | 0.02 | 0.07 | 0.1 | 0.1 | 0.23 | 0.26 |

Significance level: ** $p$ < 0.01, * $p$ < 0.05

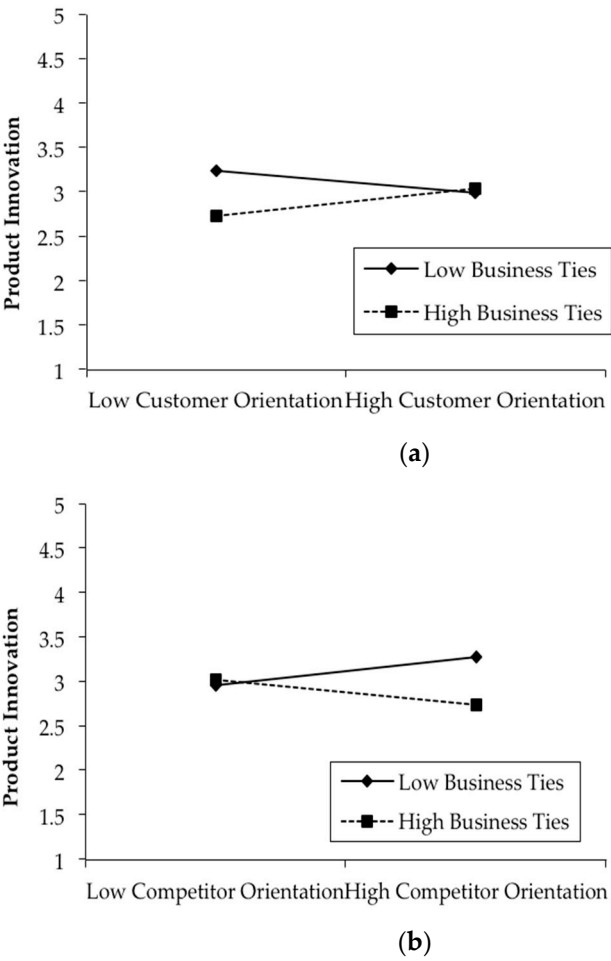

(**a**)

(**b**)

**Figure 2.** The interaction plot of (**a**) the moderating effect of business ties on the relationship between customer orientation and product innovation and (**b**) the reversed moderating effect of business ties on the relationship between customer orientation and product innovation.

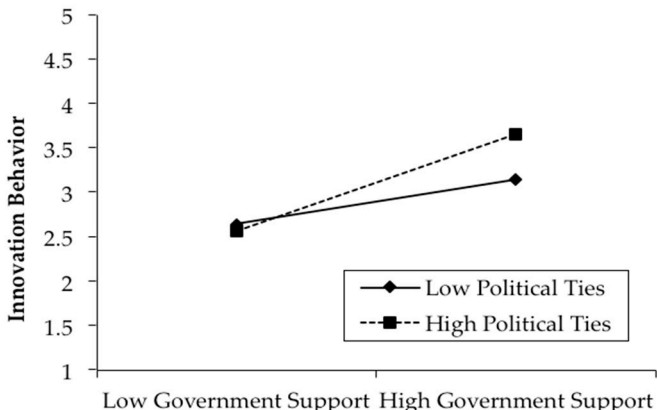

**Figure 3.** The interaction plot of the moderating effect of political ties on the relationship between government support and innovation behaviour.

**Table 3.** Results of the structural model.

| Variable | Product Innovation | Innovation Behaviour | Firm Performance |
|---|---|---|---|
| Customer Orientation | 0.13 ** (3.32) | | 0.01 (0.20) |
| Competitor Orientation | −0.01 (−0.08) | | −0.02 (−0.69) |
| Interfunctional Coordination | −0.10 ** (−2.65) | | 0.10 ** (3.39) |
| Government Support | −0.05 (−1.33) | 0.40 ** (11.72) | 0.23 ** (7.54) |
| Innovation Behaviour | 0.19 ** (4.96) | | 0.44 ** (13,45) |
| Product Innovation | | | 0.17 ** (5.88) |
| Firm Size | 0.05 (1.35) | −0.07 * (−1.95) | 0.10 ** (3.26) |
| Firm Age | −0.03 (−0.85) | 0.12 ** (3.60) | 0.01 (0.27) |

*Goodness-of-Fit Statistic:* Cmin/df = 2.36; CFI = 0.95; TLI = 0.90; NFI = 0.91; GFI = 0.98; AGFI = 0.94; RSEMA = 0.05; * $p < 0.05$, ** $p < 0.01$.

## 5. Discussion

This study addresses a gap in the research by exploring the manner in which firms can effectively use resources and capabilities, both external and internal, to enhance sustainable product innovation and sustainable innovation behaviour, which can have a significant effect on firm performance. Drawing on a new integration framework within the resource-based view, we focus on how SMEs can reinforce sustainable innovation and ensure superior firm performance. We investigated market orientation, managerial ties, and government support as tools that firms can use to recognize customers' needs, gather competitor information, and find opportunities to access rare resources. All of these can lead to positive results and solidify an organization's ability to carry out sustainable innovation. From our sample of 645 SMEs, results indicate that positive interaction between business ties and customer orientation affects product innovation. This means that when the firm has stronger business ties, it affects the firm's customer orientation and has a practical impact on product innovation. By contrast, the results also indicate that business ties negatively interact with a competitor orientation to affect product innovation, meaning that if the firm has low-level business ties that affect its competitor orientation, this will have a strongly positive impact on product innovation. Furthermore, the results show that product innovation plays a vital mediating role between customer orientation and firm performance as well as between interfunctional coordination and firm performance. The findings also indicate that the support of the government significantly indirectly affects firm performance through innovation behaviour. Moreover, the results show that firms' political ties play a considerable moderating factor, meaning that the government will more effectively support the innovation behaviour of a firm when the firm has stronger political ties. Also, innovation behaviour acts as a complete mediator, and government has an indirect relation to product innovation through innovation behaviour.

### 5.1. Theoretical Implications

Firstly, the findings revealed political and business ties as important moderator variables. The nature of these relationships can be explained by social capital theory. Results showed that market orientation and government support affect sustainable innovation development when firms establish stronger managerial ties. Therefore, cultivating relationships with an external network is an essential mission of firms, and direct connections with buyers, competitors, and suppliers can enhance the benefits of firm activities [20,53]. Findings also indicate that the interaction of business relations and customer orientation positively affects product innovation, suggesting that when the firm is highly embedded in business networks, it will receive sufficient, reliable information [24,109] to sustain innovation development. Moreover, closeness with buyers and sustained interactions with other firms can be advantageous, stimulating firms' innovation activities [110]. However, the findings also showed that the interaction of business ties and a competitor orientation negatively affects product innovation. This suggests that if a firm is too close to its business network, it might shift from responding to market needs (through new ideas and product development) to competitor operations. In other words, firms may end up relying too much on monitoring and imitating other firms instead

of committing to cutting-edge, sustainable innovations. Likewise, the findings reveal political ties as a moderator variable, with the interaction between political relations and government support positively affecting innovation behaviour. These results suggest that political ties make it easier for firms to obtain policy information or benefit from government projects [66].

This empirical study addressed a research gap by showing that the enhancement of a firm's innovation behaviour can result from a stronger political network [25,44,55,98]. Our results also shed light on the importance of government policies in ensuring the sustainability of innovation behaviour. These results are in line with theories about government policy. A firm might face constantly changing market conditions; hence, a government can define policies for supporting firms in both the short and the long terms. The role of government is to create and upgrade factors continually, such as skilled human resources, basic scientific knowledge, economic information, or infrastructure. The direct efforts of governments in the creation of these factors tend to be in generalized areas, but the most significant factors that a government can support in the interest of firm performance [2,111]—particularly in terms of innovative behaviour—are human and creative capital [82].

Secondly, we identify superior performance creation processes for firms that focus on using firm resources and capabilities. This extends the developmental resource-based literature on mediator mechanisms that explain how firms can realize and maintain superior performance [31,32,65,87]. This study employs a resource-based perspective, with a particular focus on how product innovation can serve as an essential mediating mechanism to link market orientation and firm performance. In particular, customer orientation and interfunctional coordination can influence firm performance through complete and partial mediation, respectively. In other words, our findings indicate that customer orientation and interfunctional coordination have an indirect effect on performance through product innovation. This result supports the resource-based literature that posits that the firm can base strategies on firm-specific resources. Our study shows that firms can develop superior products and achieve excellent performance under severe market competition by understanding market information from customers and then responding to customer needs by offering innovative products and services [87]. The ability to respond appropriately to customer needs and expectations is a critical resource in the development of innovative products [22,112]. Creative ideas can result from communication between employees from different departments within an organization [113]. Our study shows that when a firm has a higher level of interfunctional coordination, a lower focus on product innovation may result if the dissemination of information between departments is not for the purpose of enhancing innovation activities [23]. Consequently, this study also provides an understanding of market orientation as an essential strategy to enhance product innovation for sustainability and optimal firm performance.

The study's final theoretical implication is the expansion of the dynamic capabilities perspective. Our findings show that innovation behaviour is a link between government support and firm performance. Government involvement appears to be useful in upgrading the innovation capabilities of firms. We argue that government support can act as a critical precipitating event that triggers innovation [111]. Also, the findings indicate that innovation behaviour is fully mediating and positively affects product innovation. This result can be explained by role behaviours [114]. Sustainable product innovation requires employee capabilities and creative skills and requires that employees exhibit different role behaviours, which pertain to the individual responsibilities of jobholders in a social work environment. Also significant for a firm's sustainable innovation management is having employees with a range of personalities and attributes, such as creativity, perseverance, agreeability, and initiative [10].

*5.2. Managerial Implications*

These findings have several critical implications for owners or managers and policymakers. Firstly, owners or managers must realize the importance of interaction in their customer orientation while cultivating business ties to enhance sustainable innovation. At present, customer characteristics are changing. Customers are becoming more self-confident and active; they have access to more market

information, which enables them to make rational decisions [3]. Therefore, firms should strive to maintain consistent contact with stakeholders (customers, competitors, and suppliers) to understand their viewpoints and to create and upgrade products to better respond to consumers' needs.

Secondly, the findings showed that business ties and a competitor orientation negatively interact with product innovation. These results suggest that owners or managers who obtain strategy information and are able to assess the strengths of competitors because of a closer business network might not use this information to improve and develop their products. This would be the case if firms were only to focus on aligning their strategies with competitors or responding to the actions of competitors [24]. It is believed that competitor orientation fosters imitation of competitors and thus hampers creativity and innovation in a firm [22]. Consequently, owners and managers should be cautious about overemphasizing their business ties to avoid duplicating and imitating competitor strategies. This will help firms maintain the unique identities of their products and enhance sustainability. Firm strategies should clearly define processes and relationships and include an understanding of how to obtain stakeholder support and secure access to scarce resources to enhance sustainable innovation development. This approach can take into consideration competitors' strengths, weaknesses, and activities and can involve gathering information to benefit the firm, with the aim of making unique and creative products.

Thirdly, the findings indicate that political ties are an essential moderator connecting government policies to sustainable innovation behaviour. Government networks are central to knowledge sharing, so owners and managers should cultivate these networks to develop sustainable innovation behaviour [26,109]. Interaction with government officials and organizations can help a firm acquire timely and accurate information, access rare factors of production, and reduce the contact process necessary for developing new products in response to market demand.

Fourthly, this study suggests that owners or managers should understand the properties of product innovation as essential mediator variables that link market orientation to superior performance. The development of these resources should be a priority, and firms must understand the transmission process related to these factors. If organizations wish to survive and grow under conditions of intense competition and changing market demand, they must have innovative approaches. Furthermore, a market orientation is essential for helping managers or owners to plan sustainable business implementation through innovation that focuses on creativity and inimitability. Firms need to empathize with customers by employing customer orientation, and they should also engage in interfunctional coordination. These are critical strategies that enhance financial and market performance, because they help firms better understand the needs of target customers and how to respond to them by improving and developing products and services. If firms have these orientations, it will ensure better resource allocation and business performance. Then, firms can develop sustainable products to respond to customer needs in the long term.

Fifth, the findings suggest that innovation behaviour can serve as a mandatory mediator to enhance product innovation. Results suggest that managers and firm owners should recognize the importance of participation of employees in creating innovative products that are different from competitors. Firms need to designate a clear strategy around this, such as investing in knowledge creation and training to upgrade employees' capabilities. This could also include collaborative projects with the public sector or related organizations for the purpose of exchanging new ideas and experiences, which could inspire ideas for new, innovative products [115]. If the knowledge and competencies of employees are highly developed, they could serve as the foundation for firms' distinct competitive advantage over rivals [116]. Also, our findings suggest an indirect effect of government support on firm performance through innovation behaviour. Therefore, policies should be established that support firms' human resources development for sustainability. Government agencies should establish exchange centres and knowledge organizations for the development of employees' and entrepreneurs' sustainable innovation behaviour. For example, Thailand has a knowledge management and development organization for training entrepreneurs and employees, and southern Australia has

a Digital Tomorrow Studio, providing a place for new entrants to share ideas, concepts, and knowledge. The studio also ran workshops and invited guest speakers in response to particular expressed needs [85]. The US government supports innovation through innovation hubs, where knowledge and information about cutting edge technologies can be shared to enhance the capabilities of firms [98]. Thus, sharing ideas and learning can upgrade the sustainability behaviour of employees and entrepreneurs, and it also can help a firm with better implementation [81].

### 5.3. Limitations and Future Research

This study has a few limitations. Firstly, the questionnaire used closed questions and collected short-term data. Thus, findings may not completely reflect entrepreneurs' views and practices. A longitudinal study approach would address some of these issues and lead to other questions, as firms' development is not static but evolves over time. Secondly, this study only focused on managerial ties (i.e., business and political relations), and we do not discuss collaborative innovation networks that could enhance firm innovation. Future studies should survey collaborations between firms and customers, suppliers, competitors, universities, and research organizations to enhance sustainable innovation [117]. Thirdly, this study only investigated SMEs from Thailand. Future studies should expand this scope into other developing countries with different socioeconomic and political features and different landscapes (e.g., rural vs. urban landscapes).

**Author Contributions:** N.T. designed the conceptual framework and wrote the paper; N.T. and A.K.-H.C. performed the literature review and wrote the methods; N.T. collected and analysed the data; A.K.-H.C. rechecked the paper.

**Funding:** This research received no external funding.

**Conflicts of Interest:** The authors declare no conflict of interest.

## Appendix A

Appendix A contains a confirmatory factor analysis of the measurement model.

| Constructs and Scale Items | FL | CR | AVE |
|---|---|---|---|
| Market orientation | | | |
| *Consumer orientation (alpha = 0.93)* | | 0.93 | 0.66 |
| 1. Customer satisfaction is the primary goal of firms. | 0.98 | | |
| 2. Our strategies focus on creating superior value for customers. | 0.96 | | |
| 3. Our strategies emphasize the understanding of customer needs. | 0.50 | | |
| 4. Customer complaints and suggestions are considered essential by our firm. | 0.66 | | |
| 5. Our firm has the means to frequently measure customer satisfaction. | 0.82 | | |
| 6. We regularly analyse factors that influence the purchasing behaviour of customers. | 0.80 | | |
| 7. We have diligently traced our customers after the sale. | 0.86 | | |
| *Competitor orientation (alpha = 0.92)* | | 0.92 | 0.74 |
| 1. Our firm has shared information among departments regarding competitors' strategies. | 0.95 | | |
| 2. We emphasize fast responses to competitor actions that threaten us. | 0.79 | | |
| 3. Our firm has frequent meetings to discuss the strengths and strategies of competitors. | 0.96 | | |
| 4. We seek to anticipate the behaviour of our competitors. | 0.72 | | |
| *Interfunctional coordination (alpha = 0.95)* | | 0.95 | 0.72 |
| 1. We often communicate information about customer needs across all firm sections. | 0.70 | | |
| 2. We regularly discuss market trends across all firm sections. | 0.88 | | |
| 3. There is continual communication between departments and functions in our firm. | 0.77 | | |
| 4. Interdepartmental knowledge exchange on specific topics takes place in systematic meetings. | 0.89 | | |
| 5. Our firm distributes information on creative approaches to innovations to all firm sections. | 0.93 | | |
| 6. Discussions about our strategies occur between departments to enhance short-term and long-term goals to achieve better performance. | 0.97 | | |
| 7. Our firm continually seeks out opportunities that provide a competitive advantage. | 0.67 | | |
| Managerial ties | | | |
| *Business ties (alpha = 0.71)* | | 0.75 | 0.51 |
| The owners have been diligently using personal ties, networks, and connections with | | | |
| 1. Customers. | 0.51 | | |
| 2. Suppliers. | 0.80 | | |
| 3. Competitors. | 0.80 | | |

| | FL | CR | AVE |
|---|---|---|---|
| *Political ties (alpha = 0.89)* | | 0.89 | 0.73 |
| The owners have been diligently using personal ties, networks, and connections with | | | |
| 1. Political leaders in various levels of the government. | 0.81 | | |
| 2. Officials in industrial bureaus. | 0.85 | | |
| 3. Officials in regulatory and supporting organizations such as tax bureaus, state banks, commercial administration bureaus, and the like. | 0.90 | | |
| *Innovation behaviour (alpha = 0.88)* | | 0.91 | 0.58 |
| 1. Our personnel frequently seek out new ways to improve and develop products and services. | 0.85 | | |
| 2. Our personnel often suggest creative ideas for unique products and services. | 0.87 | | |
| 3. Our personnel have an action plan for developing new ideas. | 0.80 | | |
| 4. Our personnel are active in finding knowledge to develop skills and work behaviour in terms of innovation. | 0.87 | | |
| *Product innovation (alpha = 0.94)* | | 0.94 | 0.77 |
| 1. Customers have perceived that our products are unique and different. | 0.93 | | |
| 2. Compared to competitors' products and services, our new products and services often provide superior value for customers. | 0.93 | | |
| 3. We promptly respond to customer demands and develop new products and services. | 0.97 | | |
| 4. We continuously improve primary products and services and inject creativity into new products. | 0.73 | | |
| 5. Our firm develops new products from ideas, suggestions, or complaints that come from customers or suppliers. | 0.80 | | |
| *Government support (alpha = 0.93)* | | 0.95 | 0.80 |
| 1. Provided policies and programs that have been beneficial to firm performance. | 0.98 | | |
| 2. Provided needed knowledge and other technical support. | 0.94 | | |
| 3. Provided important market information. | 0.77 | | |
| 4. Provided external funding and financing. | 0.86 | | |
| 5. Provided information about essential regulations and helped firms obtain copyright or patent and access to rare resources. | 0.91 | | |
| *Firm performance (alpha = 0.91)* | | 0.94 | 0.76 |
| 1. Sales growth. | 0.94 | | |
| 2. Return on investment. | 0.85 | | |
| 3. Market share. | 0.75 | | |
| 4. Consumer satisfaction. | 0.95 | | |
| 5. Consumer loyalty. | 0.86 | | |

FL: Factor loading; CR: composite reliability; AVE: average of variance extracted.

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
