# Peer review of "Interactions Among Factors Influencing Product Innovation and Innovation Behaviour: Market Orientation, Managerial Ties, and Government Support"

_sustainability, doi:10.3390/su11102793_

Round 1

Reviewer 1 Report

The paper addresses an interesting research area, exploring resources and capabilities to enhance a firm’s innovation behavior and products innovation that affects a firm’s implementation. The authors draw on some prior studies to support the relationships considered in your research model. I suggest that the model should be inserted after the hypotheses development.

In my opinion the paper is very well structured and it presents a stronger discussion as well as conclusion section offers value to the reader

Author Response

Point 1: The authors draw on some prior studies to support the relationships considered in your research model. I suggest that the model should be inserted after the hypotheses development.

Response 1: I have already inserted the research model after hypotheses development.

Reviewer 2 Report

This study investigates how market orientation, managerial ties and government support affect product innovation and innovation behavior in Taiwanese SMEs operating in the creative industries. The study could be improved by addressing my comments below.

Please explain the context of the study – why focusing on creative industries? Is there anything specific about these industries? Are they manufacturing or service industries?

Conceptual framework (figure 1) should be shown in the section on literature review and hypotheses development, not in the introduction.

Please use a standard interpretation of the p-value: at the 1% level of significant, it should be p < 0.01, not p < 0.001.

Please provide detailed explanations of Figures 2 and 3.

The paper requires a thorough proof-reading, grammar checking and general use of English language. See below for some of the comments on these.

The first sentence in the introduction does not make sense: “A changing external environment, changing customer characteristics, and more sophisticated customer needs in today’s business world, firms should identify their strengths and weaknesses and devise a strategy.”

Please rewrite the second sentence in the introduction: “Strategy to cope with changing which firms must face that is innovation.”

This is not a complete sentence (introduction, page 2): Third, government support [25].

In section 4.3, this is not a complete sentence: “Therefore supporting H1a. “

Author Response

Point 1: Please explain the context of the study –

1.1 Why focusing on creative industries?

Response 1.1 The creative economy has been critical in the development of rapid world economic growth, job creation, and export expansion in the Asia region, such as China, Japan, the Republic of Korea, and Singapore, Taiwan, as well as in Thailand [1-6]. This movement also includes European regions, such as various Nordic countries, the Netherlands, and the United Kingdom [7-11. Creative industries are new opportunities for developing countries to leapfrog into emerging high-growth areas of the economy and providing a basis for national competitiveness [12].

1.2 Is there anything specific about these industries?

Response 1.2: The creative industries pay attention to intellectual capital (human, social, and structural capital). Particular, activities emphasize human creativity, ideas, skill, intellectual property, knowledge, innovation, and culture to increase the value and unique of product and service [1,2]. The driving by knowledge and creativity, chances are that economic development will be sustainably expanded [3]. These activities are difficult to duplicate by competitors. It is the core factor in achieving sustainable success firm performance [4-6].

1.3Are they manufacturing or service industries?

Response 1.3: The creative industries are creative goods and services. According to UNCTAD and UNESCO formats, creative industries are divided into four main groups and 15 branches as follows: cultural heritage (crafts, traditional medicine, food, and historical cultural tourism), arts (visual and performing arts), media (film, music, broadcasting, and printing), and functional creation (advertising, architecture, design, fashion, and software).

Point 2: Conceptual framework (figure 1) should be shown in the section on literature review and hypotheses development, not in the introduction.

Response 2: I have already moved the conceptual framework (figure 1) from the introduction section to the literature review and hypothesis development.

Point 3: Please use a standard interpretation of the p-value: at the 1% level of significant, it should be p < 0.01, not p < 0.001.

Response 3: I have already changed a standard interpretation of the p-value from p<0.001 to P < 0.01.

Point 4: Please provide detailed explanations of Figures 2 and 3.

Response 4: I have provided a more detailed explanation of Figure 2 and Figure 3. The improvement illustrates under the figure and the content in the results section.

Point5: The paper requires a thorough proof-reading, grammar checking and general use of English language. See below for some of the comments on these.

Response 5: I have improved the grammar and English language of the manuscript.

Point6: The first sentence in the introduction does not make sense: “A changing external environment, changing customer characteristics, and more sophisticated customer needs in today’s business world, firms should identify their strengths and weaknesses and devise a strategy.”

Response 6: I have already improved this sentence. The improvement has illustrated in the content of the introduction section. The improving sentence as “A changing external environment, changing customer characteristics, and more sophisticated customer needs have made it increasingly hard for firms to retain competitiveness. Some entrepreneurs have struggled to compete and succeeded, while others have failed. Therefore, entrepreneurs and firms should develop effective strategies to cope in the current environment [1–3]”

Point7: Please rewrite the second sentence in the introduction: “Strategy to cope with changing which firms must face that is innovation.”

Response 7: I have already rewritten this sentence. The improvement has illustrated in the content of the introduction section. The improving sentence as “Innovation is an important weapon that firms can use to safeguard their existence, and it can give firms a competitive advantage over rivals”

Point8: This is not a complete sentence (introduction, page 2): Third, government support [25].

Response 8: I have already improved this sentence. The improvement has illustrated in the content of the introduction section. The improving sentence as “Third, support from the government sector is also important”

Point9: In section 4.3, this is not a complete sentence: Therefore supporting H1a.

Response 9: I have already improved this sentence. The improvement has illustrated in the content of the results section. The improving sentence as “ So H1a was supported”

Round 2

Reviewer 2 Report

My comments are addressed at a satisfactory level.

Author Response

Point 1: English language and style

( ) Extensive editing of English language and style required
(x) Moderate English changes required
( ) English language and style are fine/minor spell check required
( ) I don't feel qualified to judge about the English language and style

Response 1: The article has been English changes by a native speaker in order to improve the writing.
